# Construction of N−E bonds via Lewis acid-promoted functionalization of chromium-dinitrogen complexes

Zhu-Bao Yin [ORCID], Gao-Xiang Wang, Xuechao Yan, Junnian Wei ✉ & Zhenfeng Xi [ORCID] ✉

Direct conversion of dinitrogen ($N_2$) into N-containing compounds beyond ammonia under ambient conditions remains a longstanding challenge. Herein, we present a Lewis acid-promoted strategy for diverse nitrogen-element bonds formation from $N_2$ using chromium dinitrogen complex [Cp*($I^iPr_2Me_2$) Cr($N_2$)$_2$]K (**1**). With the help of Lewis acids $AlMe_3$ and $BF_3$, we successfully trap a series of fleeting diazenido intermediates and synthesize value-added compounds containing N−B, N−Ge, and N−P bonds with 3 d metals, offering a method for isolating unstable intermediates. Furthermore, the formation of N−C bonds is realized under more accessible conditions that avoid undesired side reactions. DFT calculations reveal that Lewis acids enhance the participation of dinitrogen units in the frontier orbitals, thereby promoting electrophilic functionalization. Moreover, Lewis acid replacement and a base-induced end-on to side-on switch of [NNMe] unit in [(Cp*($I^iPr_2Me_2$)CrNN(BEt$_3$)(Me)] (**8**) are achieved.

The current utilization of dinitrogen ($N_2$) predominantly relies on the Haber-Bosch process, also known as ammonia ($NH_3$) synthesis[1–3]. Over the past six decades, the N−H bond formation facilitated by homogenous transition metal catalysts via associative and dissociative pathways has been extensively studied[4–13]. Typical pathways for synthesizing $NH_3$, such as the Chatt[14] and Schrock[15] cycles, along with studies by Nishibayashi[16] and Peters[17], have been successively proposed. However, developing methods for the direct dinitrogen-element (N−E) bond formation beyond $NH_3$ lags behind and remains a longstanding and challenging issue[18–38]. The formation of diazenido and hydrazido intermediates in associative pathways is pivotal for constructing N−E bonds, but two major scientific challenges remain unresolved. Firstly, diazenido intermediates formed via the initial electrophilic functionalization of coordinated $N_2$ tend to decompose, especially with 3 d metals, through N−H and N−Si homolytic cleavage, β-silyl elimination or other unclear pathways (Fig. 1a)[26,29,30,39–42]. Secondly, the thermolabile nature of methyl-, silyl-, germyl-, and phosphanyldiazenes complicates the isolation of diazenido compounds containing N−C, N−Si, N−Ge, and N−P bonds (Fig. 1a)[43–45]. To address the above issues, we draw inspiration from nitrogenase, which features a multi-metallic active site surrounded by crucial amino acid residues that polarize $N_2$ and enhance charge transfer from the iron center to $N_2$[46–48]. Using both transition-metal complexes and Lewis acids (LA) to co-activate $N_2$ thus represents a promising approach[18–25,28,49–58], but further electrophilic derivatizations of the coordinated $N_2$ using Lewis acids remain extremely limited[22,28,54]. In this work, we present a Lewis acid-promoted strategy for forming N−C, N−Si, N−Ge, and N−P bonds. More accessible conditions for N−C bond formation were achieved by inhibiting undesired side reactions[59] at the Cr(0) center (Fig. 1b, top). Furthermore, we use Lewis acids to trap and stabilize a series of fleeting diazenido intermediates containing N−B, N−Si, N−Ge, and N−P bonds (Fig. 1b, below), confirming the feasibility of the initial step of electrophilic derivatization of Cr-$N_2$ complexes to form various N−E bond-containing diazenido complexes.

## Results

### Synthesis and characterization

Considering that the $N_\beta$ atom in diazenido intermediates is already $sp^2$-hybridized with a lone pair of electrons[60,61], we hypothesized that additional Lewis acids might trap and stabilize such intermediates

Beijing National Laboratory for Molecular Sciences (BNLMS), Key Laboratory of Bioorganic Chemistry and Molecular Engineering of Ministry of Education, College of Chemistry, Peking University, Beijing 100871, China. ✉e-mail: jnwei@pku.edu.cn; zfxi@pku.edu.cn

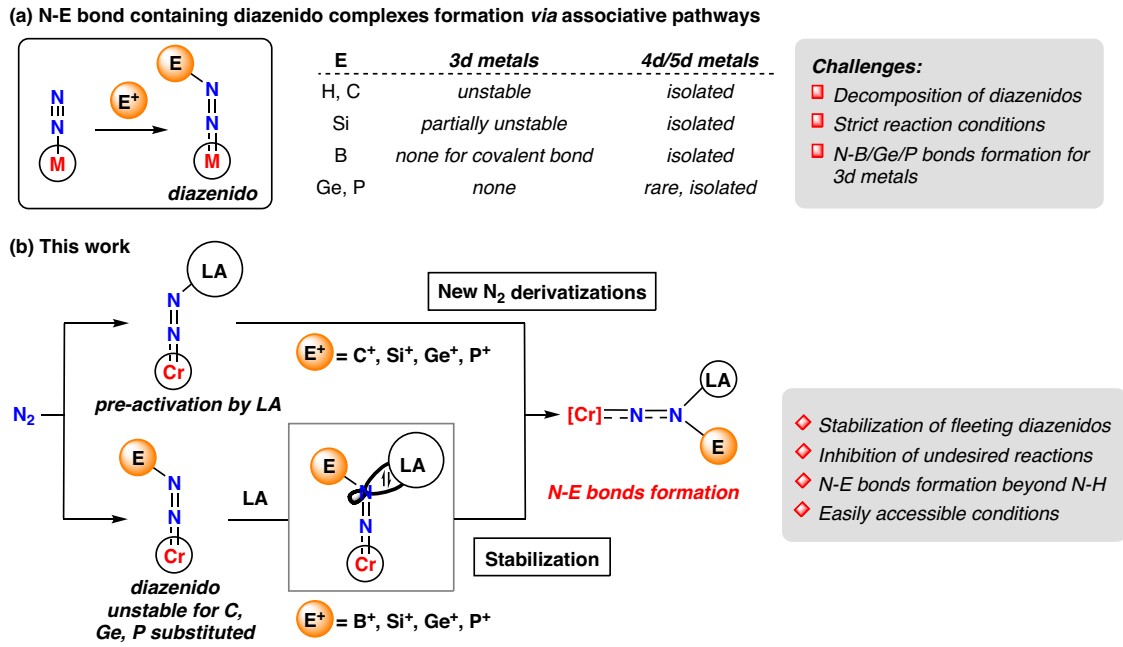

**Fig. 1 | N−E bond formation via associative pathways. a** Challenges for diverse N−E bonds formation via diazenido complexes. **b** N−E bond formation beyond N−H bond promoted by Lewis acids. LA Lewis acid.

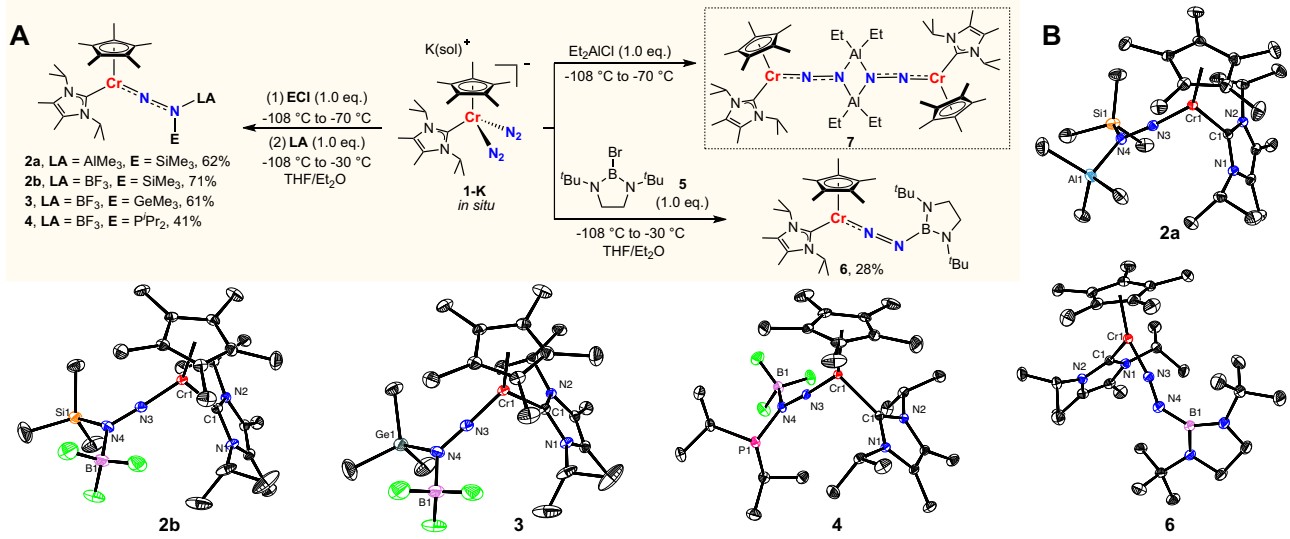

**Fig. 2 | Synthesis and molecular structures of complexes 2, 3, 4, 6 and 7.**
**A** Synthesis of complexes **2a**, **2b**, **3**, **4**, **6** and **7**. **B** Molecular structure of **2a**, **2b**, **3**, **4**, and **6** with thermal ellipsoids at 30% probability. H atoms omitted for clarity. Selected bond lengths [Å] and angles [deg] of **2a**: Cr1-N3 1.6801(3), N3–N4 1.3060(17), N4–Al1 1.9659(17), N3–N4–Si1 111.98(11); **2b**: Cr1-N3 1.6768(13), N3–N4 1.3108(17), N4–B1 1.574(2), N3–N4–Si1 115.34(10); **3**: Cr1-N3 1.684(2), N3–N4 1.295(3), N4–B1 1.574(4), N3–N4–Ge1 116.27(18); **4**: Cr1-N3 1.6753(11), N3–N4 1.3144(16), N4–B1 1.581(2), N3–N4–P1 113.47(10); **6**: Cr1-N3 1.7104(17), N3–N4 1.231(2), N4–B1 1.423(3), N3–N4–B1 138.56(19).

(Fig. 1b, below). As given in Fig. 2A, this idea was validated by adding 1.0 equiv of AlMe₃ or BF₃•Et₂O to the proposed fleeting diazenido intermediates [Cp*(I^iPr₂Me₂)Cr(NNSiMe₃)] (formed in situ by reaction of [Cp*(I^iPr₂Me₂)Cr(N₂)₂]K(sol) **1-K** and 1.0 equiv of Me₃SiCl, sol = THF or Et₂O, I^iPr₂Me₂ = 1,3-diisopropyl-4,5-dimethylimidazol-2-yildene, Fig. 2A, left) at low temperature, resulting in the Lewis acid trapped products **2a** and **2b**. Both **2a** and **2b** are paramagnetic and have a solution magnetic moment of 2.4(1) $\mu_B$ and 2.7(1) $\mu_B$ at 296 K, respectively. X-ray crystallography reveals that compared with the diazenido [Cp*(I^iPr₂Me₂)Cr(NNSi^iPr₃)] (1.243(2) Å for N−N bond)[35], the N−N distances in **2a** and **2b** are significantly elongated, with lengths of 1.3060(17) Å and 1.3108(17) Å, respectively (Fig. 2B).

It should be pointed out that N−Ge or N−P bond-containing complexes cannot be synthesized from **1-K** because undesired redox reactions always arise during the direct functionalization of **1-K** with Me₃GeCl or ^iPr₂PCl, leading primarily to [Cp*(I^iPr₂Me₂)CrCl] and [Cp*(I^iPr₂Me₂)Cr(η¹-N₂)(μ-η¹:η¹-N₂)Cr(I^iPr₂Me₂)Cp*]. To our delight, the method used to prepare **2a** and **2b** provides an opportunity to isolate Lewis acid-stabilized species **3** and **4** (Fig. 2A, left). Both **3** and **4** are paramagnetic and have a solution magnetic moment of 2.4(1) $\mu_B$ and 2.7(1) $\mu_B$ at 296 K, respectively. The N−N bond lengths of **3** (1.295(3) Å) and **4** (1.3144(16) Å) also lie between the values for a N−N single bond (1.46 Å for H₂NNH₂) and a N = N double bond (1.25 Å for HN = NH) (Fig. 2B). These results indicate that the Lewis acids can

effectively trap fleeting diazenido intermediates and prevent undesired side reactions.

B and Al-based electrophiles, unlike their Si, Ge, and P counterparts, possess an extra empty p orbital, making them capable of acting as Lewis acids. Therefore, when reacting with $N_2$ complexes, the products often display a dinuclear structure, as seen in forms such as $[(MNNBR_2)_2]^{23}$ and $[(MNNAlR_2)_2]^{36,50}$. In fact, the dinuclear complex **7** can be obtained during the reaction of complex **1-K** with 1.0 equiv $Et_2AlCl$ (Fig. 2A, right), although obtaining high-quality data for **7** has been challenging. The reaction between **1-K** and common B-based electrophiles ($Cy_2BCl$, $Ph_2BCl$, $Mes_2BF$) does not result in boryldiazenido intermediates, but instead leads to side reactions, such as single electron transfer or the dissociation of $I^iPr_2Me_2$. Fortunately, treatment of **1-K** with the more electron-rich boron electrophile **5** yields the boryl-functionalized diazenido complex **6** (Fig. 2A, right). Compound **6** is paramagnetic and has a solution magnetic moment of 2.8(1) $\mu_B$ at 296 K. The Cr1–N3 bond length of crystal **6** (1.7104(17) Å) is shorter than the typical single bond, indicating a multiple bond characteristic. The N–N bond distance (1.231(2) Å) is in the range of the typical N=N double bond, and the N3–N4–B1 angle (138.56(19)°) suggests an $sp^2$ hybridized geometry for the $N_\beta$ atom. The N4–B1 bond length (1.423(3) Å) is shorter than complexes **2b, 3**, and **4**, indicating N4–B1 bond is a covalent bond with multiple B–N bonding (Fig. 2B). The presence of an empty p orbital on the B atom, along with the significant steric hindrance, are crucial for stabilizing and isolating this diazenido product. The strong vibration peak at 1602 cm$^{-1}$ in IR spectra of **6** is assigned to N–N vibration of the $\eta^1$-diazenide fragment. A $^{15}N$–$^{15}N$ stretching vibration at 1540 cm$^{-1}$ of the $^{15}N_2$-labeled sample of **6** is consistent with the mass difference between $^{15}N_2$ and $^{14}N_2$ (Fig. S6).

To elucidate how Lewis acids stabilize those Cr diazenido intermediates, **2b** was analyzed using density functional theory (DFT) calculations as an example. Significant energy release ($\Delta G = -24.6$ kcal/mol) from the corresponding Cr diazenido and $BF_3 \cdot THF$ to **2b** indicates a strong driving force for the Lewis acid coordination and the enhanced stability of complex **2b**. Moreover, Fuzzy bond order (FBO) analysis shows that the FBO of N–N bonds decreases from 1.9 to 1.6 after $BF_3$ coordination, accompanied by Cr–N bond shortening (Fig. 3), consistent with the strong activation effect of Lewis acid on diazenido complexes[62]. Mayer bond orders and Wiberg bond indices (WBI) are also provided (see Fig. S41 for details), and the trends are qualitatively consistent.

We have recently reported several $N_2$ functionalization reactions utilizing mono/bis-phosphine or NHC-appended cyclopentadienyl Cr–$N_2$ complexes, yielding N–H, N–C, N–Si bonds, and hetero-bimetallic Cr–$N_2$ complexes[29,33,35,64]. However, setbacks arose when employing mono-phosphine or NHC-appended cyclopentadienyl Cr–$N_2$ complexes to investigate N–C bond formation. Direct functionalization of complex **1-K** with MeOTf, MeOTs, $Me_3OBF_4$, or MeI did not yield N–C bond products. In addition, our aforementioned reaction processes given on the left of Fig. 2A involving the reaction of **1-K** with MeOTf followed by adding Lewis Acid $BEt_3$ produced only very few crystals of the N–C bond formation product $[(Cp^*(I^iPr_2Me_2)CrNN(BEt_3)(Me)]$ (**8**). Hence, we speculated that an alternative strategy would be needed for making N–C bonds. After experimenting, we found that the order of adding **LA** (Lewis acids) and **EX** (electrophiles) had a remarkable effect on the N–C bond formation. Thus, treating **1-crypt** with 1.0 equiv of $BEt_3$ or $AlMe_3$ led to two new vibration peaks (Fig. 4C, left, red line at 1738 cm$^{-1}$, 1900 cm$^{-1}$ for $BEt_3$ and blue line at 1755 cm$^{-1}$, 1911 cm$^{-1}$ for $AlMe_3$) accompanied by incomplete conversion of **1-crypt**. These two new peaks are assigned as $N_2$-related peaks because the corresponding $^{15}N$ peaks with $BEt_3$ were at 1685 cm$^{-1}$ and 1838 cm$^{-1}$ (for details, see Figure S10), suggesting an equilibrium between **1-crypt** and the Lewis acids adducts may exist. The reduced $N_2$ stretching, indicative of lower bond order and increased polarization, is reminiscent of the effect of acidic residues in nitrogenase active sites on Fe-bound $N_2$, which enhance polarization and facilitate protonation. This inspired further addition of MeOTf to explore N–C bond formation. Fortunately, complex **8** was successfully isolated this time (Fig. 4A). Complex **8** is paramagnetic and has a solution magnetic moment of 3.2(1) $\mu_B$ at 296 K. The N–N bond length of crystal **8** is 1.289(2) Å, reflecting the strong interaction between the empty boron p-orbital and the $N_2$ $\pi^*$ orbital. The N3–N4–B1 angle is 124.01(17)°, suggesting $sp^2$ hybridization of the $N_\beta$ atom with a dative coordinated boron atom (Fig. 4B). Furthermore, we found that complexes **2a, 2b, 3**, and **4** could also be synthesized using this method.

To investigate the role of Lewis acids in the above-discussed N–C bond formation, we conducted experiments to verify the interaction between $BEt_3$ and **1-crypt**. When we added 2.0 equiv of $BEt_3$ to the solution of **1-crypt**, the peaks at 1738 cm$^{-1}$ and 1900 cm$^{-1}$ were significantly enhanced (see Fig. 4C, right, red line for 1.0 equiv $BEt_3$ and green line for 2.0 equiv $BEt_3$). This suggests an equilibrium between **1-crypt** and **10-**$BEt_3$, with **1-crypt** favoring reaction with two equivalents of $BEt_3$ (Fig. 4A). The coupling vibration of two $[NNBEt_3]$ substituents shifts one peak to lower frequencies and the other to higher

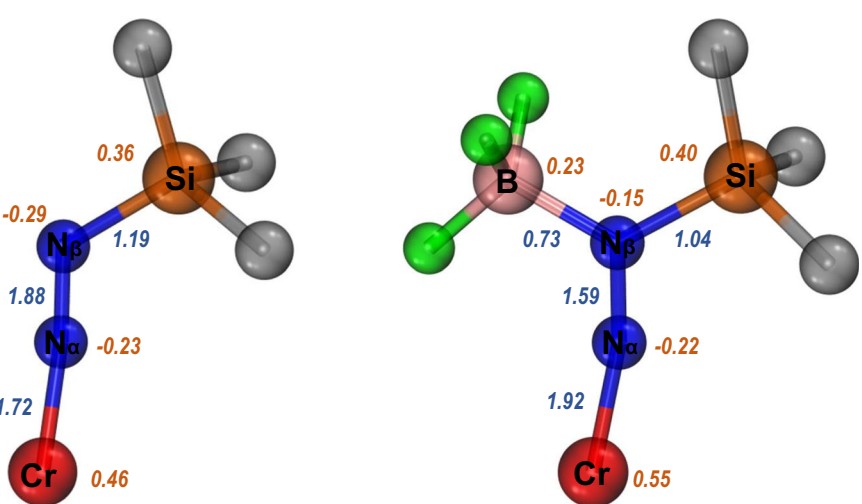

**Fig. 3 | DFT calculations.** Fuzzy bond orders (blue) and Hirshfeld atomic charges (orange) for $[Cp^*(I^iPr_2Me_2)Cr(NNSiMe_3)]$ (left) and **2b** (right). Wavefunctions were generated at the PWPB95/def2-QZVPP level of theory.

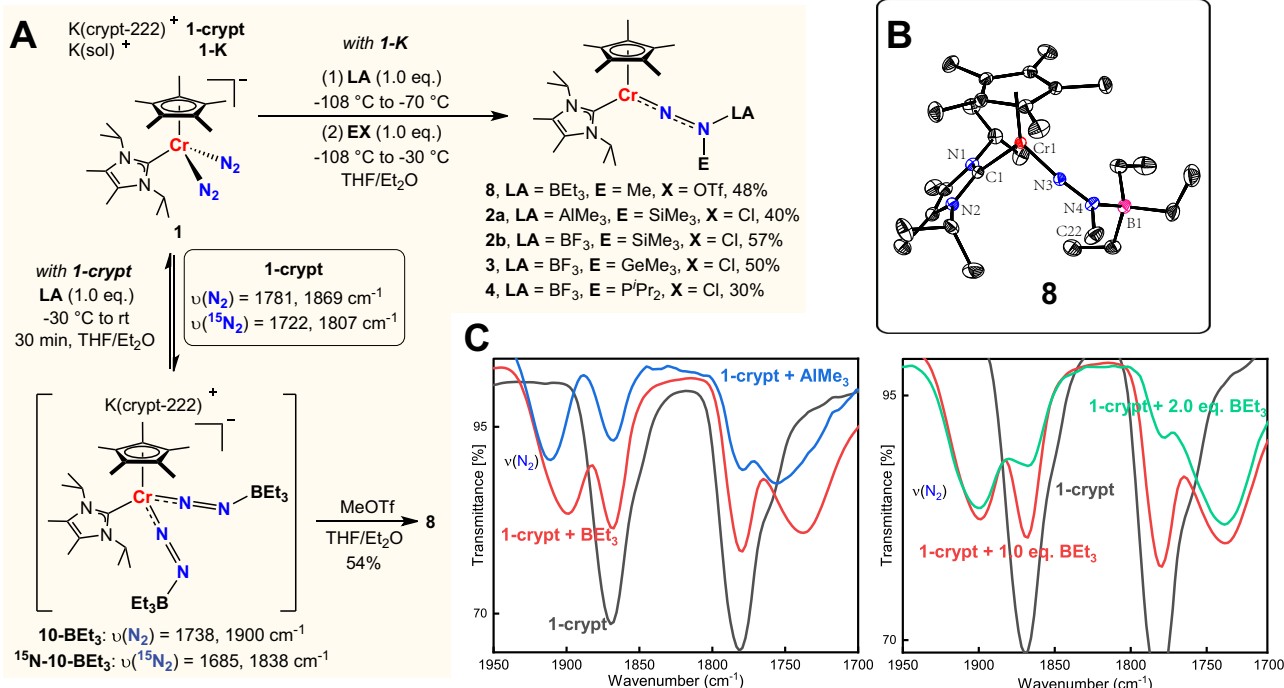

**Fig. 4 | Synthesis and mechanistic investigations. A** Synthesis of complexes **2a, 2b, 3, 4,** and **8. B** Molecular structure of **8** with thermal ellipsoids at 30% probability. H atoms omitted for clarity. Selected bond lengths [Å] and angles [deg] of **8**:

Cr1−N3 1.6902(16), N3−N4 1.289(2), N4−B1 1.618(3), N3−N4−C22 114.19(18). **C** Variation of vibration peaks during adding Lewis acids to **1·crypt**.

frequencies. In addition, two new peaks (385 and 416 ppm), distinct from **1·crypt** (404 and 413 ppm), were observed in the in situ $^{15}$N NMR spectra of **1·crypt** with 2.0 equiv. BEt$_3$. The $\Delta\delta$ for Cr-bound and terminal N atoms increases from 9 to 31 ppm, indicating polarization of N$_2$ unit but less than Szymczak's Fe-N$_2$ system ($\Delta\delta$ = 109 ppm)[54]. This result is consistent with smaller IR shifts (1780, 1869 cm$^{-1}$ to 1738, 1900 cm$^{-1}$). The calculated stretching vibration peaks for the NN bond in the IR spectrum are 1779 cm$^{-1}$ and 1878 cm$^{-1}$ for **1·crypt**, and 1752 cm$^{-1}$ and 1917 cm$^{-1}$ for **10·BEt$_3$**, which are in good agreement with experimental values. DFT calculations show that the $\Delta G$ value for **1·crypt** coordination with one equivalent of BEt$_3$ in THF at −30 °C is +0.7 kcal/mol, whereas coordination with two equivalents is −0.8 kcal/mol, confirming the equilibrium between **1·crypt** and **10·BEt$_3$**. AIM (atoms in molecules) analysis of **10·BEt$_3$** confirms dative B−N interactions (see Figure S42 for details). Moreover, BEt$_3$ coordination significantly enhances the participation of the N$_2$ moiety in HOMO (Orbitals contributions in HOMO: N$_\alpha$ 13.8%, N$_\beta$ 20.2%, Cr 34.5% for **10·BEt$_3$** and N$_\alpha$ 7.1%, N$_\beta$ 12.0%, Cr 46.1% for **1·crypt**, see Figure S43 for more details). This enhancement increases the likelihood of NN unit participation in functionalization and reduces the side reactions between Cr and electrophilic reagents.

Interestingly, coordination of two BEt$_3$ molecules lowers the activation barrier for functionalization with MeOTf (18.7 to 15.6 kcal/mol), while it increases with one BEt$_3$ (18.7 to 22.3 kcal/mol), consistent with the higher yield of **8** using 2.0 equiv. BEt$_3$. Therefore, the primary role of BEt$_3$ is to mitigate undesirable side reactions between MeOTf and the Cr center to some extent. Furthermore, once the N−C bond is formed, BEt$_3$ does not dissociate, stabilizing the CrNNMe framework and preventing its rapid decomposition. Diazenido intermediates stabilized by Lewis acid coordination exhibit characteristic UV-Vis absorption peaks around 620 nm (Figures S23−S29). Due to the structural similarity of these complexes, **2b** was chosen as a representative for TD-DFT calculations to simulate its UV-Vis spectrum. NTO analysis reveals that the absorption peak around 620 nm primarily arises from π to π* excitation within the CrNN unit (Figure S44).

Further transformations of these Lewis acid-coordinated products were explored. However, the reactions of **2b** with CO$_2$, $^t$BuNCO, PhSiH$_3$, $^n$Bu$_3$SnH, DIBAL-H or KC$_8$ did not yield any clear N-containing products yet. Encouragingly, the coordinated BEt$_3$ in **8** can initiate two distinct reaction pathways with a Lewis acid or base. First, BEt$_3$ can be replaced by AlMe$_3$, forming **9**, with no significant structural changes observed after Lewis acid substitution (Fig. 5). Second, the removal of BEt$_3$ by IMe$_4$ (1,3-dimethyl-4,5-dimethylimidazol-2-ylidene) induces a transition of the [NNMe] moiety from end-on to side-on coordination, yielding the bis-side-on [NNMe]-coordinated complex **11** (Fig. 5). The N−N distance increases from 1.289(2) Å to 1.387(2) and 1.389(2) Å, indicating a greater potential for further conversion of the [NNMe] fragment.

Functionalization of N$_2$ with electrophiles often requires strict low-temperature conditions to prevent undesired side reactions[29]. Given its success in evading undesired redox reactions to form N−E bonds, we persisted in investigating the potential of more accessible reaction conditions. UV−Vis measurements revealed that two bands (463 nm, 654 nm) attributed to complex **8** emerged when 1.0 equiv BEt$_3$ and 1.0 equiv MeOTf were added to **1·crypt** successively at −30 °C, 0 °C and 20 °C (for details, see Figures S30−S32). Based on these results, we conducted a 0.05 mmol scale reaction at approximately room temperature, confirming the formation of **8**, albeit with difficulty in determining the yield.

In summary, we have synthesized a series of N−E (E = B, C, Si, Ge, P) bond-containing complexes from N$_2$ with the aid of Lewis acids, offering a potential approach for isolating N−B, N−Ge, N−P bond-containing compounds in 3 d metals. Extra Lewis acids not only suppress undesirable side reactions, but also provide possibilities for more accessible reaction conditions. Moreover, an end-on to side-on switch of [NNMe] unit is achieved, creating greater opportunities for N$_2$ transformations. We hope this study will facilitate the development of more manageable and diverse N$_2$ functionalization reactions.

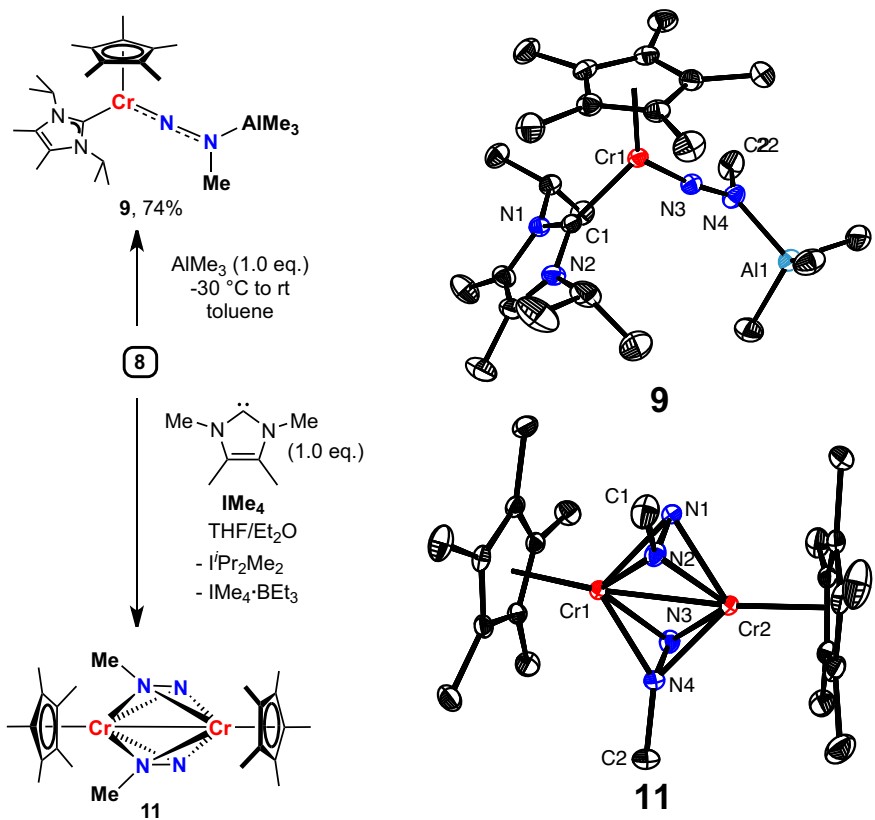

**Fig. 5 | Further transformation of complex 8.** Selected bond lengths [Å] and angles [deg] of **9**: Cr1−N3 1.694(2), N3−N4 1.286(3), N4−Al1 1.944(2), N3−N4−Al1 121.28(16); **11**: N1−N2 1.389(2), N3−N4 1.387(2), N1−N2−C1 119.39(15), N3−N4−C2 119.42(17).

## Methods

### General procedure for the synthesis of Lewis acids stabilization complexes

**Conditions A**. In a nitrogen atmosphere glovebox, excess $KC_8$ (0.3 mmol, 40.5 mg) was added into the THF (6 mL) solution of complex **Cp\*Cr(I$^i$Pr$_2$Me$_2$)Cl** (0.1 mmol, 40.3 mg). The solution was stirred for 24 h at room temperature, generating the Cr(0)-$N_2$ complex **1-K**, as evidenced by two peaks (1760 cm$^{-1}$, 1846 cm$^{-1}$) in the IR spectra. The solvent was filtered, and the filtrate was concentrated to approximately 4 mL before adding 2 mL of Et$_2$O. The solution was frozen in the coldwell chilled externally with liquid nitrogen. Meanwhile, a solution of Lewis acids (0.1 mmol) in hexane or Et$_2$O was also frozen in the coldwell chilled externally with liquid nitrogen. Immediately upon thawing, the solution of Lewis acids was added to the frozen **1-K** equipped with a magnetic stirring bar. The solution was slowly warmed to −70 °C while stirring for 30 min. Then the reaction solution was frozen in the coldwell chilled externally with liquid nitrogen again. Meanwhile, a solution of electrophile (**EX**) (0.1 mmol) in THF was also frozen in the coldwell chilled externally with liquid nitrogen. Immediately upon thawing, the solution of electrophile was added to the frozen reaction solution. The solution was slowly warmed to −30 °C while stirring for 70 min. Volatile materials were removed under vacuum. The solid residues were extracted with hexane/Et$_2$O or Et$_2$O/ THF. The filtrate was concentrated and placed in a −30 °C freezer, yielding crystals.

**Conditions B**. In a nitrogen atmosphere glovebox, excess $KC_8$ (0.3 mmol, 40.5 mg) was added into the THF (6 mL) solution of complex **Cp\*Cr(I$^i$Pr$_2$Me$_2$)Cl** (0.1 mmol, 40.3 mg). The solution was stirred for 24 h at room temperature, generating the Cr(0)-$N_2$ complex **1-K**, as evidenced by two peaks (1760 cm$^{-1}$, 1846 cm$^{-1}$) in the IR spectra. The solvent was filtered, and the filtrate was concentrated to

approximately 4 mL before adding 2 mL of Et$_2$O. The solution was frozen in the coldwell chilled externally with liquid nitrogen. Meanwhile, a solution of electrophile (**EX**) (0.1 mmol) in THF was also frozen in the coldwell chilled externally with liquid nitrogen. Immediately upon thawing, the solution of electrophile was added to the frozen **1-K** equipped with a magnetic stirring bar. The solution was slowly warmed to −70 °C while stirring for 30 min. Then the reaction solution was frozen in the coldwell chilled externally with liquid nitrogen again. Meanwhile, a solution of Lewis acids (0.1 mmol) in hexane or Et$_2$O was also frozen in the coldwell chilled externally with liquid nitrogen. Immediately upon thawing, the solution of Lewis acids was added to the frozen reaction solution. The solution was slowly warmed to −30 °C while stirring for 70 min. Volatile materials were removed under vacuum. The solid residues were extracted with hexane/Et$_2$O or Et$_2$O/ THF. The filtrate was concentrated and placed in a −30 °C freezer, yielding crystals.

**Computational details**. Density functional theory (DFT) calculations were performed using ORCA 6.0.0 to investigate the electronic structures[65]. All geometric structures were optimized at the TPSSh/ def-TZVP level of theory[66], incorporating dispersion corrections via the Becke-Johnson damping scheme (D3BJ)[67]. The optimized geometries closely match the single-crystal structures, supporting the validity of the computational approach. Additionally, to further ensure accuracy, we conducted single-point energy calculations on the optimized geometries using the double-hybrid functional PWPB95 with def2-QZVPP basis sets[68]. Solvent effects were considered by employing the SMD implicit solvent model with tetrahydrofuran (THF) as the solvent in these single-point calculations[69]. UV-Vis spectrum of **2b** was computed using the long-range-corrected DFT functional CAM-B3LYP[70] with the def2-TZVP basis sets based on the optimized geometric structure using the Gaussian 16 package[71]. We have carefully validated the spin

states of all paramagnetic species, ensuring that the computed electronic structures are physically meaningful. The figures were prepared by Visual Molecular Dynamics (VMD) program[72], and the corresponding wavefunction analysis was performed by Multiwfn[73].

## Data availability

Crystallographic data for the structures reported in this article have been deposited at the Cambridge Crystallographic Data Centre under deposition numbers CCDC-2356388 (**2a**), CCDC-2356389 (**2b**), CCDC-2356390 (**3**), CCDC-2356392 (**4**), CCDC-2356391 (**6**), CCDC-2356394 (**8**), CCDC-2356393 (**9**), CCDC-2405607 (**11**) are available from CCDC in cif format. These data can be obtained free of charge from The Cambridge Crystallographic Data Centre via www.ccdc.cam.ac.uk/data_request/cif. The optimized computational structures are provided separately as a "Source Data.xlsx" file. All other data supporting the findings of this study are available within the article and its Supplementary Information, or from the corresponding author upon request. Source Data containing optimized Cartesian coordinates are provided in this paper. Source data are provided with this paper.

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

## Acknowledgements

This work was supported by the National Natural Science Foundation of China (no. 21988101, Z.X.) and the Postdoctoral Fellowship Program of CPSF under Grant Number GZB20230014 (Z.-B.Y.). The DFT calculation was supported by the High-performance Computing Platform of Peking University. The authors thank Dr. Haihan Yan for his kind help and discussions.

## Author contributions

Z.-B.Y., J.W. and Z.X. conceived the work and designed the experiments. Z.-B.Y. performed synthetic, spectroscopic, X-ray and mechanistic investigations work. Z.-B.Y. performed the crystallographic data analyses. J.W. performed the computational work. Z.-B.Y., X.Y. carried out the low-temperature UV–vis spectroscopy. Z.-B.Y., G.-X.W., X.Y., J.W. and Z.X. discussed the results in detail and commented on the manuscript.

## Competing interests

The authors declare no competing interests.
