## [Transparent Peer Review file · Nature Communications]

Construction of N–E bonds via Lewis acid-promoted functionalization of chromium-dinitrogen complexes

Corresponding Author: Dr Junnian Wei

Version 0:

Reviewer comments:

Reviewer #1

(Remarks to the Author)

In this work, Yin, Wang, Yan, Wei, and Xi present a novel Lewis acid-promoted strategy for constructing nitrogen-element (N–E) bonds from dinitrogen (N₂) using a chromium-dinitrogen complex. The authors employ DFT calculations to elucidate key electronic structure aspects of their systems, with a focus on the stabilisation of fleeting diazenido intermediates by Lewis acids. While the work is innovative and addresses a challenging problem, making it potentially suitable for publication in a high-impact journal such as Nature Communications, the computational analysis is underdeveloped and needs substantial improvements before the manuscript can be considered for acceptance. Below, I provide specific comments and recommendations regarding the theoretical analysis.

1 - The authors have optimised their structures using the r2SCAN-3c method. While r2SCAN-3c is adequate for initial geometry optimisations due to its efficiency, it is not sufficient for final geometries, especially for paramagnetic systems involving transition metals like chromium. The current computational approach lacks the necessary accuracy for such complex systems. The authors should redo their computations using higher-level, state-of-the-art DFT functionals with inclusion of dispersion corrections. Triple-zeta basis sets are required for transition metals, while double-zeta basis sets can be acceptable for lighter elements. Additionally, the authors must ensure proper assignment of spin states and that spin contamination is minimised throughout their analysis.

2 - The Computational Methods section in the SI is insufficient. The authors should provide detailed information about the methods used and justify their choice. Appropriate references should be included for each computational tool or method applied. Moreover, merely presenting MOs without a proper discussion is inadequate. The authors need to provide a qualitative and quantitative analysis of the MOs, along with their implications for the reactivity of the chromium-dinitrogen complexes. Furthermore, the computational analysis should be expanded to include additional species.

3 - The manuscript does not provide sufficient detail about how atomic contributions to the MOs were calculated. The authors should clearly describe the method used.

4 - It is unclear whether the authors have accounted for solvent effects in their ΔG calculations. Given that reactions with associative/dissociative steps are particularly sensitive to solvation, the authors should incorporate implicit solvent models to obtain more realistic energy values. Additionally, they should consider including concentration corrections in their ΔG calculations.

5 - The computational analysis does not address any aspect of the reaction mechanism. The authors should provide more computational information about the mechanistic pathway, including energy profiles and the identification of key intermediates and transition states.

6 - The authors selected Fuzzy bond order analysis to investigate bond order, but no justification is provided for this choice. FBO is one of several bond order methods available in Multiwfn, and it is important to explain why FBO was chosen over other methods. To ensure consistency and robustness in bond order analysis, the authors should compare values obtained from different methods and present them in a table in the SI. This analysis should not be performed using the current computational approach, as the level of theory needs to be improved to ensure reliable bond order estimates.

7 - While the authors describe the effect of Lewis acid coordination on the bonding of N₂, more detailed bonding analysis should be performed. Several established methods are available in the literature and could provide deeper insights into how the electronic structure of N₂ changes upon coordination with Lewis acids. The authors should select and apply the method of their choice to elucidate these bonding changes.

8 - In Figure 4C, the authors discuss the variation of vibrational peaks during the addition of Lewis acids to 1-crypt. This experimental data would benefit from a thorough computational vibrational analysis. The authors should compute the vibrational spectra for the key intermediates and complexes, and compare the calculated vibrational frequencies with the experimental data.

9 - The authors provide some UV-vis experimental data, but computational characterisation of these spectra is missing. Long-range-corrected DFT functionals should be used to compute the UV-vis spectra, as these functionals are more accurate for electronic excitations involving charge-transfer transitions. Natural Transition Orbital (NTO) analysis should be performed to examine the nature of the electronic transitions and to correlate the computed spectra with experimental observations.

Reviewer #2

(Remarks to the Author)
Please see attachment.

Reviewer #3

(Remarks to the Author)

The manuscript of Wei, Xi and coworkers presents a new approach to electrophilic functionalization of TM-bound N₂ molecules, by the initial coordination of Lewis acids, followed by the electrophile. This is demonstrated on a paramagnetic CpCr(NHC) framework, and results in the construction of bonds from nitrogen to a range of p-block elements.

To me, the work is very nice and the N-E bonds are impressive, but it seems a little incomplete, as it is really just the first steps in a more full N₂ functionalization / dissociation. I would be OK with accepting the manuscript with its current chemistry, but further functionalization of the N₂ unit would give a more complete picture, while also showing something closer to real utility. Adding something like that would be a great improvement, and would make the manuscript a better fit in a prestigious journal such as this.

Some more minor points:

- Fig 1 is very confusing - the variable E is defined underneath the word "diazenido", but then E⁺ is defined just to the right of that, with different atoms. This is duplication, but also they don't match entirely, which is confusing. LA should also be defined as Lewis acid.

- Fig 2 - above the arrow, "E⁺" is stated as a reagent. However, E⁺ itself is not added to the reaction. The actual reagent should be specified (E-Cl or E-X, where E = SiMe₃, ... ; X = ...). The same goes for Fig 4.

- The synthesis and structure of 7 should be shown somewhere in a figure.

- The manuscript needs to be more explicit about whether the compounds are paramagnetic or not. For some complexes it is stated, for others it is missing (e.g. 3 and 4).

- I assume the paramagnetism is also the reason for the missing C, B, F and N NMR data, but it would be nice if the authors would state this clearly somewhere in the manuscript, and if they attempted to record spectra for these other nuclei.

- Conclusion: "offering a viable approach for isolating N-B, N-Ge, N-P bond-containing value-added compounds in 3d metals for the first time" - this is a bit of an exaggeration. I wouldn't consider it a viable approach to isolating value-added compounds, because no real value-added compounds were prepared in the work. However, it could be a "potential" approach to such compounds.

Reviewer #4

(Remarks to the Author)

Version 1:

Reviewer comments:

Reviewer #1

(Remarks to the Author)

The authors have made significant efforts to enhance the theoretical analysis and have addressed all my concerns thoroughly and satisfactorily. I am pleased to recommend this manuscript for publication and congratulate the authors for their work.

Reviewer #2

(Remarks to the Author)

The authors have revised their manuscript and addressed most of the points raised by the reviewers, especially by providing more computational and experimental data, greatly improving its quality. Initially, I was skeptical about what the authors claimed for the formulation of 10-BEt3 as a bis-adduct based on their IR data. However, they successfully solidified their characterisation and their claim by providing theoretical N-N IR stretches and energy profiles, and experiments with an excess of the LA. Authors also proposed the design of a new species, compound 11, by extrusion of BEt3 in complex 8 with a Lewis base (IMe4). This species is quite uncommon (dimerization phenomenon and leaving of the NHC ligand from Cr) and is a nice example of how Lewis acids may help access original N2-activation complexes. If my initial feeling was that the manuscript was more fitted for a specialised journal, I acknowledge that the authors have seriously taken into account the criticisms of all reviewers and strengthened substantially the chemistry described here. My appreciation of this paper in terms of novelty does not change, but if laudatory reports come back from the two other reviewers and that the editor gives acceptance as final decision, I congratulate to the authors for having this elegant work published.

Reviewer #3

(Remarks to the Author)

The authors have impressively tackled the Herculean task of addressing all of the 30+ comments of referees 1&2, although I will leave it up to these referees to say how well the points were addressed. My comments were dealt with satisfactorily, and the addition of further reaction chemistry adds some weight to the work. I am happy to recommend acceptance, with two minor further points:

- Complex 11 seems to be quite novel to me. Is there anything like it in the literature? Is it the first example of a complex with side-on-bound alkylated or substituted N2 ligands? This could be explored and, if it is, mentioned.

- Complex 11 is not named in the text.

Reviewer #4

(Remarks to the Author)

Point-by-Point Response to Comments of Reviewers

Response to Comments of Reviewer 1

Reviewer #1 (Remarks to the Author):

In this work, Yin, Wang, Yan, Wei, and Xi present a novel Lewis acid-promoted strategy for constructing nitrogen-element (N–E) bonds from dinitrogen (N₂) using a chromium-dinitrogen complex. The authors employ DFT calculations to elucidate key electronic structure aspects of their systems, with a focus on the stabilisation of fleeting diazenido intermediates by Lewis acids. While the work is innovative and addresses a challenging problem, making it potentially suitable for publication in a high-impact journal such as Nature Communications, the computational analysis is underdeveloped and needs substantial improvements before the manuscript can be considered for acceptance. Below, I provide specific comments and recommendations regarding the theoretical analysis.

Response: We thank the reviewer for the valuable comments. Followings, please find our replies to all your comments. We have significantly improved the manuscript by adding new content, which has substantially enhanced the overall quality of the work.

1) Reviewer 1 wrote: The authors have optimised their structures using the r2SCAN-3c method. While r2SCAN-3c is adequate for initial geometry optimisations due to its efficiency, it is not sufficient for final geometries, especially for paramagnetic systems involving transition metals like chromium. The current computational approach lacks the necessary accuracy for such complex systems. The authors should redo their computations using higher-level, state-of-the-art DFT functionals with inclusion of dispersion corrections. Triple-zeta basis sets are required for transition metals, while double-zeta basis sets can be acceptable for lighter elements. Additionally, the authors must ensure proper assignment of spin states and that spin contamination is minimised throughout their analysis.

Response: We sincerely thank the reviewer for their valuable feedback. In response to the suggestion, we have re-optimized all structures using ORCA 6.0.0 with the TPSSh functional and triple-zeta basis sets (def-TZVP), incorporating dispersion corrections via the Becke-Johnson damping scheme (D3BJ). Additionally, to further ensure accuracy, we conducted single-point energy calculations on the optimized geometries using the double-hybrid functional PWPB95 with def2-QZVPP basis sets. We have also carefully verified the correct assignment of spin states for all complexes, ensuring that the computed electronic structures are physically meaningful.

2) Reviewer 1 wrote: The Computational Methods section in the SI is insufficient. The authors should provide detailed information about the methods used and justify their choice. Appropriate references should be included for each computational tool or method applied. Moreover, merely presenting MOs without a proper discussion is inadequate. The authors need to provide a qualitative and quantitative analysis of the MOs, along with their implications for the reactivity of the chromium-dinitrogen complexes. Furthermore, the computational analysis should be expanded to include additional species.

Response: We appreciate the reviewer's valuable feedback. Detailed descriptions of all

computational methods have been provided.

In addition, we have expanded the analysis of molecular orbitals (MOs) in the main text. And additional species relevant to the paper have also been included in the analysis.

Figure S43 HOMO of anion part of **10-BEt₃** (left) and **1-crypt** (right). Contributions were determined using wavefunctions generated by the double-hybrid functional PWPB95/def2-QZVPP and analyzed through the Hirshfeld method, as implemented in Multiwfn.

The discussion “Moreover, BEt₃ coordination significantly enhances the participation of the N₂ moiety in HOMO (Orbitals contributions in HOMO: N_α 13.8%, N_β 20.2%, Cr 34.5% for **10-BEt₃** and N_α 7.1%, N_β 12.0%, Cr 46.1% for **1-crypt**, see Figure S43 for more details). This enhancement increases the likelihood of NN unit participation in functionalization and reduces the side reactions between Cr and electrophilic reagents.” has been added to the main text.

However, as this work is primarily focused on the novel functionalization of chromium-dinitrogen complexes, with the synthetic innovation being the key contribution, we have selected only representative complexes for analysis to ensure that the computational discussion does not overshadow the main experimental findings.

3) Reviewer 1 wrote: The manuscript does not provide sufficient detail about how atomic contributions to the MOs were calculated. The authors should clearly describe the method used.

Response: Thank your for highlighting this point. In response, we have provided details in the manuscript regarding the calculation of atomic contributions to the MOs (Figure S43). Specifically, these contributions were determined using wavefunctions generated by the double-hybrid functional PWPB95/def2-QZVPP and analyzed through the Hirshfeld method, as implemented in Multiwfn. All relevant information has been added to both the main text and the Supporting Information.

4) Reviewer 1 wrote: It is unclear whether the authors have accounted for solvent effects in their ΔG calculations. Given that reactions with associative/dissociative steps are particularly sensitive to solvation, the authors should incorporate implicit solvent models to obtain more realistic energy values. Additionally, they should consider including concentration corrections in their ΔG calculations.

Response: Thank you for this important observation. In the revised manuscript, we have incorporated solvent effects by applying the SMD implicit solvent model with tetrahydrofuran (THF) as the solvent in single-point energy calculations using the PWPB95/def2-QZVPP level of theory. For the ΔG calculations, we used single-point energies from PWPB95/def2-QZVPP combined with vibrational analysis results from TPSSh/def-TZVP at a reaction temperature of $-30\text{ }^{\circ}\text{C}$. To better deal with low frequencies, we applied a quasi-RRHO treatment, with entropy interpolation between the harmonic oscillator and free-rotor approximations, (by *Shermo* program, *Comput. Theor. Chem.* **2021**, *1200*, 113249)

Additionally, we have considered concentration corrections in our ΔG calculations. The initial ΔG values were calculated under ideal gas conditions, which correspond to a solution-phase standard state of approximately 0.04 M — closely matching the actual concentration of our reaction system.

5) Reviewer 1 wrote: The computational analysis does not address any aspect of the reaction mechanism. The authors should provide more computational information about the mechanistic pathway, including energy profiles and the identification of key intermediates and transition states.

Response: In the revised manuscript, we have included a detailed discussion of the reaction mechanism, specifically examining the activation energy changes in the reaction between complex **1-K** and MeOTf with the coordination of the Lewis acid BEt₃. We also discussed the effect of Lewis acid coordination on the transition state energies, providing a more comprehensive mechanistic analysis.

Table S9 Calculated energies (in Hartree). The solvent model was applied during the single-point energy calculations. **Note:** The default geometry optimization includes THF as a solvent using the SMD model. However, for complex **2b**, there is always a small imaginary frequency when the solvent model is applied during the optimization, despite several attempts to eliminate it. Therefore, for complex **2b** and the corresponding thermodynamic data calculations, the geometry optimization was performed without the solvent model.

	G (TPSSh/def- TZVP)	G-E (TPSSh/def- TZVP)	Single point energy (PWPB95/def2- QZVPP)	G corrected by Shermo at $-30\text{ }^{\circ}\text{C}$
geometry optimized without solvent model				
BF ₃	-324.698941	-0.013159	-324.593301	-324.600382

MeOTf	-1001.552152	0.031234	-1001.376399	-1001.338315
BF ₃ •Et ₂ O	-558.363659	0.115721	-558.226877	-558.102941
Et ₂ O	-233.659946	0.106346	-233.601615	-233.488626
Cp*(I ⁺ Pr ₂ Me ₂)Cr(NNSiMe ₃)	-2494.208773	0.571054	-2493.772434	-2493.194048
2b	-2818.918886	0.583241	-2818.439346	-2817.847603
geometry optimized with THF as the solvent				
BEt ₃	-262.505823	0.163990	-262.430888	-262.262105
MeOTf	-1001.560774	0.030993	-1001.376860	-1001.338994
1-crypt anion	-2457.127109	0.666885	-2456.696831	-2193.762070
TS 1-crypt anion with MeOTf	-3196.150357	0.528566	-3195.609204	-3195.071290
1-crypt coordinate 1.0 eq. BEt ₃	-2457.127109	0.666885	-2456.696831	-2456.023038
TS 1-crypt coordinate 1.0 eq. BEt ₃ with MeOTf	-3458.658763	0.722694	-3458.058249	-3457.326439
1-crypt coordinate 2.0 eq. BEt ₃	-2719.624872	0.857715	-2719.152250	-2718.287510
TS 1-crypt coordinate 2.0 eq. BEt ₃ with MeOTf	-3721.156986	0.913362	-3720.523690	-3719.601183

Calculations indicate that the activation energy increases if only one BEt₃ coordinates to the dinitrogen ligand, whereas coordination of two BEt₃ molecules to the dinitrogen significantly lowers the activation barrier for the reaction with MeOTf. Furthermore, the calculations suggest that the reaction can proceed even in the absence of BEt₃, indicating that the primary role of BEt₃ is to substantially stabilize the unstable diazenido intermediate. These discussions have been incorporated into the main text of the revised manuscript.

6) Reviewer 1 wrote: The authors selected Fuzzy bond order analysis to investigate bond order, but no justification is provided for this choice. FBO is one of several bond order methods available in Multiwfn, and it is important to explain why FBO was chosen over other methods. To ensure consistency and robustness in bond order analysis, the authors should compare values obtained from different methods and present them in a table in the SI. This analysis should not be performed using the current computational approach, as the level of theory needs to be improved to ensure reliable bond order estimates.

Response: In line with our previous publications, we chose Fuzzy bond order (FBO) analysis because it has proven to be a robust method for describing bond orders in transition metal–dinitrogen complexes, correlating well with Lewis structures and chemical intuition.

However, following the reviewer’s recommendation, we have also included Mayer bond orders and Wiberg bond indices (WBI) for comparison, with all values presented in Figure S41 in the Supporting Information (SI). These analyses were conducted based on the wavefunction generated from single-point energy calculations at the PWPB95/def2-QZVPP level of theory.

Figure S41 Comparison of Fuzzy bond order, Mayer bond order and Wiberg bond order for $[\text{Cp}^*(\text{I}^*\text{Pr}_2\text{Me}_2)\text{Cr}(\text{NNSiMe}_3)]$ (left) and **2b** (right). Wavefunctions were generated using the PWPB95/def2-QZVPP level of theory.

7) Reviewer 1 wrote: While the authors describe the effect of Lewis acid coordination on the bonding of N_2 , more detailed bonding analysis should be performed. Several established methods are available in the literature and could provide deeper insights into how the electronic structure of N_2 changes upon coordination with Lewis acids. The authors should select and apply the method of their choice to elucidate these bonding changes.

Response: Thanks for your suggestion, we conducted a more detailed bonding analysis using **1-crypt** coordinated with BEt_3 as an example. Specifically, we examined the changes in HOMO orbital contributions and the Bond Orders of the Cr–N and $\text{N}\equiv\text{N}$ bonds, revealing the role of Lewis acid (Figure S43).

Figure S43 HOMO of anion part of **10-BEt₃** (left) and **1-crypt** (right). Contributions were determined using wavefunctions generated by the double-hybrid functional PWPB95/def2-QZVPP and analyzed through the Hirshfeld method, as implemented in Multiwfn.

We then performed an Atoms in Molecules (AIM) analysis for the dinitrogen complex with two equivalents of BEt_3 (**10-BEt₃**). This analysis identified (3, -1) bond critical points (BCPs), marked as blue dots in Figure S42. The low electron density values at the B–N (3, -1) BCP, along with positive Laplacian values ($\nabla^2\rho(r)$), indicate that the B–N interactions are dative bonds. Additionally, to further investigate the electron distribution in complex **10-BEt₃**, we generated a Localized Orbital Locator (LOL) map, which provides a real-space visualization of electron (de)localization, providing a real-space visualization of electron (de)localization. The LOL map further illustrates the dative bond nature of the B–N bond.

Figure S42 The plot of localized orbital locator (LOL) map of the **10-BEt₃**. (Notes: This analysis identified (3, -1) bond critical points (BCPs), marked as blue dots. The low electron density values at the B–N (3, -1) BCP, along with positive Laplacian values ($\nabla^2\rho(r)$), indicate that the B–N interactions are dative bonds. The localized orbital locator (LOL) map, which provides a real-space visualization of electron (de)localization, providing a real-space visualization of electron (de)localization, further illustrates the dative bond nature of the B–N bond.) Wavefunctions were generated using the PWPB95/def2-QZVPP level of theory.

8) Reviewer 1 wrote: In Figure 4C, the authors discuss the variation of vibrational peaks during the addition of Lewis acids to **1-crypt**. This experimental data would benefit from a thorough computational vibrational analysis. The authors should compute the vibrational spectra for the key intermediates and complexes, and compare the calculated vibrational frequencies with the experimental data.

Response: We performed re-optimizations and frequency calculations using the TPSSh/def-TZVP level of theory. Based on the *Database of Frequency Scale Factors for Electronic Model Chemistries* (*J. Chem. Theory Comput.* **2010**, *6*, 2872-2887), we applied a frequency scaling factor of 0.96. The computed vibrational frequencies for the dinitrogen ligand in **1-crypt** were 1878 cm^{-1} and 1779 cm^{-1} , which closely matched the experimental values. Upon coordination with two equivalents of BEt_3 , the calculated infrared vibrational peaks shifted to 1917 cm^{-1} and 1752 cm^{-1} , also aligning well with the experimental data, supporting the formation of a bis-coordinated species upon Lewis acid coordination.

Additionally, we calculated the ΔG values for the coordination of one and two equivalents of

BEt₃ to the **1-crypt** anion in THF at -30 °C, obtaining values of +0.7 kcal/mol for one equivalent and -0.8 kcal/mol for two equivalents. These findings further confirm that bis-coordination is more favorable and indicate the presence of a dynamic equilibrium in solution.

9) Reviewer 1 wrote: The authors provide some UV-vis experimental data, but computational characterisation of these spectra is missing. Long-range-corrected DFT functionals should be used to compute the UV-vis spectra, as these functionals are more accurate for electronic excitations involving charge-transfer transitions. Natural Transition Orbital (NTO) analysis should be performed to examine the nature of the electronic transitions and to correlate the computed spectra with experimental observations.

Response: Thank you for the suggestion. Given the structural similarity of the Lewis acid-stabilized diazenido complexes, we performed UV-vis spectral calculations on a representative complex, **2b**. We employed the long-range-corrected DFT functional CAM-B3LYP with the def2-TZVP basis set in a TD-DFT calculation. We have incorporated solvent effects by applying the SMD implicit solvent model with THF as the solvent.

In the computed UV-vis spectrum, the characteristic peak around 620 nm corresponds to the S₀→S₁ transition. NTO analysis revealed that this absorption is primarily dominated from the π to π^* excitation of the Cr=N=N unit. The relevant calculated data and NTO visualizations have been included as Figure S44 in the Supporting Information.

Figure S44 UV-Vis spectrum of **2b** computed using the long-range-corrected DFT functional CAM-B3LYP with the def2-TZVP basis set in a TD-DFT calculation. (Notes: Solvent effects were incorporated by applying the SMD implicit solvent model with THF as the solvent. In the computed UV-vis spectrum, the characteristic peak around 620 nm corresponds to the S₀→S₁ transition. NTO analysis revealed that this absorption is primarily dominated by the π to π^*

excitation within the Cr=N=N unit.)

Response to Comments of Reviewer 2

Reviewer #2 (Remarks to the Author):

This article by Yin *et al.* studies (by spectroscopic, crystallographic, and computational methods) the influence of Lewis acids (LA) to the distal N of Cr-N₂ complexes on derivatization of the N₂ ligand into a silyl-, germyl-, boryl, aluminyl-, phosphanyl- or methyl-diazenido one, by reaction with main-group electrophiles. Their results points to a positive effect of such association, granting isolation of a family of N₂-derived diazenido compounds, in particular the methylsubstituted ones that could not be prepared without LA association. This manuscript also highlights two reactional approaches to get these fleeting diazenido compounds: i) Reaction of [Cp*(L)Cr(N₂)₂]K with the LA to trigger preactivation of the terminal nitrogen of the N₂ unit which is then subjected to electrophilic functionalization with a Si; Ge, C, or P-based electrophilic agent. ii) The reverse approach, which entails the *in situ* generation of the reactive fleeting diazenido intermediates - [Cp*(L)Cr(μ -N₂-E)] - that is then treated with the LA to isolate and trap the resulting species. This set of results is indeed important in the context of N fixation beyond ammonia and further establishes transition metal complexes as the best way to prepare value-added nitrogenous compounds from N₂. Formation of diazenido compounds from N₂ is not new, but, as the authors underline, difficult with 3d metals. Facilitating this by introduction of LA coordination is clever. Such a strategy builds on the seminal work from the Szymczak group demonstrating that for iron-N₂ complexes, adjunction of an LA that coordinates to the terminal N of the N₂ ligand enhances back-donation from the metal, making the N₂ ligand more electron-rich at the terminal N and allowing selective electrophilic functionalization thereof, instead of reactivity at the metal center. This work is technically sound, the species are well characterized and I congratulate the authors for managing handling of such low-valent Cr-N₂ complexes that are very challenging sensitive species, some successfully trapped and isolated at low temperature (down to -100 °C). They are comprehensively characterized by a wide range of tools. In my opinion, this work represents a nice follow-up of this group's chemistry on Cr-N₂ complexes, but lacks novelty in terms of N₂ transformation. It is rather an elegant combination of different known reactivity of N₂ complexes that lead the authors to solve a problem on a reaction (the treatment of **1** or related complexes with electrophiles) they have already reported on (JACS 2019, JACS 2023, JACS 2023). Therefore, I consider it would be more suited for a specialized journal. Would they choose to submit elsewhere or would the other reviewers be less severe so that the manuscript has chances to make its way through, I invite the authors to consider addressing the following points which, I believe, would greatly improve the quality of the manuscript and lift some interrogations.

Response: We thank the reviewer for valuable comments. Based on the suggestions provided by you and other reviewers, we have significantly improved the manuscript by adding new content, which has substantially enhanced the overall quality of the work.

In particular, we have discovered that BEt₃ can be removed by IMe₄ (IMe₄ = 1,3-dimethyl-4,5-

dimethylimidazol-2-ylidene), leading to a rare and novel transition of the [NNMe] moiety from end-on to side-on coordination. This represents an interesting and previously unexplored transformation. This part was added in the main text as Fig. 5.

Fig. 5 | Further transformation of complex 8. Selected bond lengths [Å] and angles [deg] of **9**: Cr1–N3 1.694(2), N3–N4 1.286(3), N4–Al1 1.944(2), N3–N4–Al1 121.28(16); **11**: N1–N2 1.389(2), N3–N4 1.387(2), N1–N2–C1 119.39(15), N3–N4–C2 119.42(17).

The transformation of N₂ has always been our goal, we actually have tried our best to explore numerous possibilities for further converting these products. However, the reactions of **2b** with CO₂, ^tBuNCO, PhSiH₃, ⁿBu₃SnH, DIBAL-H or KC₈, as well as the anticipated [3+2] cycloaddition of **4** with CO₂, ^tBuNCO, or RC≡CR (R = Me, COOMe) have not yet succeeded. Gratifyingly, the coordinated BEt₃ in **8** can be replaced by the more acidic AlMe₃, forming **9**, with no significant structural changes observed after Lewis acid substitution (Fig. 5).

Reviewer 2 wrote: “Over the past six decades, the N–H bond formation facilitated by transition metals via associative and dissociative pathways has been extensively studied” It is not clear what the authors mean by associative and dissociative pathways. Do they refer to alternating and distal pathways? When saying transition metals, do they only consider soluble complexes? “N–H bond formation” is vague and could refer to a lot of transformations, while given the content of the article it is likely that the authors refer specifically to N₂ and N₂ derived ligands. Please rephrase.

Response: We thank for the valuable suggestion from the reviewer.

- In dinitrogen activation, the dissociative pathway involves the cleavage of the N≡N bond before further reactions while the associative pathway proceeds through stepwise activation of N₂, forming intermediates like diazenido and hydrazido without full bond dissociation. Moreover, the associative pathways can be further divided into distal pathway and alternating pathway according to the the reaction site (N_α or N_β) of the second step derivatization.
- In this manuscript, we focus on dinitrogen conversion reactions in homogeneous systems, so the transition metals here refer to homogeneous catalysts.
- “N–H bond formation” in this manuscript refer specifically to N₂ derived ligands (diazenido, hydrazido, hydrazine, imido, amido, amine) or the final NH₃ or NH₄⁺

products.

We have provided clarifications and made revisions in the main text.

Reviewer 2 wrote: “the isolation of diazenido compounds containing N-C, N-Si, N-Ge, and N-P bonds” Reference 41 is given to support the words, the part 1 of Wiberg’s review about azenes is more appropriate (part 2 is cited): Wiberg, N. Silyl, Germyl, and Stannyl Derivatives of Azenes, N_nH_n : Part I. Derivatives of Diazene, N_2H_2 . in *Advances in Organometallic Chemistry* (eds. Stone, F. G. A. & West, R.) vol. 23 131–191 (Academic Press, 1984). A more recent example of isolation of poorly stable silyldiazene adducts of Lewis acids is also worth mentioning: Reiß, F., Schulz, A. & Villinger, A. Synthesis, Structure, and Reactivity of Diazene Adducts: Isolation of iso-Diazene Stabilized as a Borane Adduct. *Chem. Eur. J.* 20, 11800–11811 (2014).

Response: We changed the reference 41 to “Wiberg, N. Silyl, germyl, and stannyl derivatives of azenes, N_nH_n part I. derivatives of diazene, N_2H_2 . *Adv. Organomet. Chem.* 23, 131–191 (1984).”

And we added “*Chem. Eur. J.* 20, 11800–11811 (2014)” as reference 45.

Reviewer 2 wrote: “Using both transition-metal complexes and Lewis acids (LA) to co-activate N_2 thus represents a promising approach” Beyond the references given by the authors, other chemists have reviewed this topic: Ruddy, A. J., Ould, D. M. C., Newman, P. D. & Melen, R. L. Push and pull: the potential role of boron in N_2 activation. *Dalton Trans.* 47, 10377–10381 (2018); Simonneau, A. & Etienne, M. Enhanced Activation of Coordinated Dinitrogen with p-Block Lewis Acids. *Chem. Eur. J.* 24, 12458–12463 (2018). Some recent development should also be added: Jori, N. et al. Iron promoted end-on dinitrogen-bridging in heterobimetallic complexes of uranium and lanthanides. *Chem. Sci.* 15, 6842–6852 (2024); Escomel, L. et al. Coordination of $Al(C_6F_5)_3$ vs. $B(C_6F_5)_3$ on group 6 end-on dinitrogen complexes: chemical and structural divergences. *Chem. Sci.* 15, 11321–11336 (2024).

Response: The above references were added.

Reviewer 2 wrote: “More accessible conditions for N-C bond formation were achieved by inhibiting undesired side reactions at the Cr(0) center” It would be interesting to note here that for an iron-phosphine- N_2 complex, electrophiles such as H^+ or Me^+ reacted at the metal while Si^+ electrophiles reacted at the N_2 ligand, similar to the authors’ Cr complexes: Field, L. D., Hazari, N. & Li, H. L. Nitrogen Fixation Revisited on Iron(0) Dinitrogen Phosphine Complexes. *Inorg. Chem.* 54, 4768–4776 (2015).

Response: We added the above reference for comparison and better understanding.

Reviewer 2 wrote: “the N_β atom in diazenido intermediates is already sp^2 -hybridized with a lone pair of electrons” It is probably pertinent to cite here the reviews by Sutton and Dilworth about diazenido compounds: Dilworth, J. R. Diazene, diazenido, isodiazene and hydrazido complexes. *Coord. Chem. Rev.* 330, 53–94 (2017); Sutton, Derek. Organometallic diazo compounds. *Chem. Rev.* 93, 995–1022 (1993).

Response: Added.

Reviewer 2 wrote: “Both **2a** and **2b** are paramagnetic and have a solution magnetic moment of 2.4(1) μ_B and 2.7(1) μ_B at 296 K, respectively” Are these values consistent with the expected spin state of Cr(II) within a strong field, half-sandwich coordination?

Response: Yes, the solution magnetic moment of 2.4(1) μ_B and 2.7(1) μ_B indicate that $S = 1$ spin state of **2a** and **2b**, which is consistent with their X-ray structures and DFT calculations.

Reviewer 2 wrote: “The reaction between **1-K** and common B-based electrophiles (Cy₂BCl, Ph₂BCl, Mes₂BF) fail to construct N–B bonds, likely due to insufficient steric hindrance” The meaning of this sentence is not clear. Cy and Mes group are rather bulky. The steric hindrance should play a role if a N–B bond is formed, yet the authors declare that this was not the case. Could the authors clarify whether they could evidence N–B bond formation (e.g., by IR) and that the thus formed boryldiazenido compound is unstable, or if decomposition occurs without N–B bond formation and in that case the argument of steric hindrance is not adequate?

Response: We thank the reviewer for the valuable comments. We have carried out reactions between **1-K** and Mes₂BF, Ph₂BCl, Cy₂BCl again. The Cr(I)-N₂ complex was detected by IR when Mes₂BF or Ph₂BCl were used as electrophiles, indicating that a SET (single electron transfer) process occurred between **1-K** and these reagents.

In addition, the formation of I'Pr₂Me₂•Mes₂BF was observed when Mes₂BF was used as the electrophile.

Based on these results, we propose three possible reaction pathways between **1-K** and alkyl or aryl substituted B-based electrophiles: (1) a direct SET process between **1-K** and R₂BX; (2) the formation of boryldiazenidos, which may decompose easily into Cr(I)-N₂ complex and/or I'Pr₂Me₂•R₂BX; (3) the alkyl or aryl substituted B-based electrophiles may directly destroy the I'Pr₂Me₂ ligand to give I'Pr₂Me₂•R₂BX.

Accordingly, we have revised the main text of the manuscript. “The reaction between **1-K** and common B-based electrophiles (Cy₂BCl, Ph₂BCl, Mes₂BF) does not result in borydiazenido intermediates, but instead leads to side reactions, such as single electron transfer or the dissociation of $I^*Pr_2Me_2$. Fortunately, treatment of **1-K** with the more electron-rich boron electrophile **5** yields the boryl-functionalized diazenido complex **6**.”

Reviewer 2 wrote: “and less electrophilicity B-based electrophile” should read “and less electrophilic B-based electrophile”

Response: Revised.

Reviewer 2 wrote: “and the N3-N4-B1 angle (138.56(19)°) suggests an sp² hybridized geometry for the N_β atom.” This angle significantly deviates from the 120° expected for an ideal sp²-hybridized p-block atom. Can the authors explain this and compare whether this is the case for other boryldiazenido compounds?

Response: We thank the reviewer for the valuable comments. We found that this phenomenon is common in boryldiazenido compounds. For example, in Mo and W boryldiazenido complexes [(dppe)₂MNNB(C₆F₅)₂]⁺ (M = Mo, W) (*Angew.Chem. Int. Ed.* **56**, 12268 (2017)) the bond angles of the N-N-B are 140.45° and 138.42°, respectively. We believe this may be due to the presence of empty p-orbital on boryl groups. The N–B bond lengths (about 1.4 Å) indicate that N–B bond is a covalent bond with multiple B–N interactions, which leads to a deviation from the standard sp² hybridized geometry.

Reviewer 2 wrote: surprisingly, the authors do not comment on the success or failure of applying their second strategy (adding LA to **1** then E⁺) to the problematic boron halides Cy₂BCl, Ph₂BCl and Mes₂BF. Have the authors tried?

Response: Thank you for your comment. Actually, both the first and second strategies were indeed tested for constructing N–B bonds.

(1) As shown in the following figure, when the first strategy was applied with **5** as the electrophile, complex **6** was still the final product without the need for additional Lewis acid. This confirms that the N–B bond in complex **6** is a covalent bond involving multiple B–N interactions, which means that the boron atom with an empty orbital itself is an additional Lewis acid.

(2) When the second strategy was used with Cy₂BCl, **5** as electrophiles, BF₃•Et₂O, AlMe₃ as Lewis acids, we did not obtain specific results corresponding to the desired dinitrogen conversion. We hypothesize that these electrophiles may react with the I^tPr₂Me₂ ligand.

Reviewer 2 wrote: can the authors assign N-N stretching frequency in their IR characterizations of the diazenido complexes (or at least part of them) with the help of ¹⁵N-labeling?

Response: The vibration peaks of boryldiazenido complexes ¹⁴N-**6** and ¹⁵N-**6** are located at 1602 cm⁻¹ and 1540 cm⁻¹, respectively, which is consistent with the mass difference between ¹⁵N₂ and ¹⁴N₂.

The corresponding information has been added in SI as Figure S6. “A ¹⁵N–¹⁵N stretching vibration at 1540 cm⁻¹ of the ¹⁵N₂-labeled sample of **6** is consistent with the mass difference between ¹⁵N₂ and ¹⁴N₂ (Figure S6).”

Reviewer 2 wrote: “This phenomenon, reminiscent of Lewis acids mimicking nitrogenase” is in my opinion too vague to express what the authors mean here. It should be rephrased in that sense: “The lowering of the stretching frequency for the N_2 ligand is indicative of a diminished bond order and greater polarization. This is reminiscent of the influence of acidic residues within the active site of the nitrogenase on Fe-bound N_2 , further polarizing it and presumably assisting protonation.” Reference to the work of Szymczak (ref. 51) must be made here.

Response: The sentence was revised to “The reduced N_2 stretching, indicative of lower bond order and increased polarization, is reminiscent of the effect of acidic residues in nitrogenase active sites on Fe-bound N_2 , which enhance polarization and facilitate protonation. This inspired further addition of MeOTf to explore N–C bond formation.”.

Reviewer 2 wrote: “Complex **8** is paramagnetic and has a solution magnetic moment of 3.2(1) μB at 296 K” Same as above, does this value allows to extract a meaningful spin state?

Response: The solution magnetic moment of 3.2(1) μ_B indicates a $S = 1$ spin state for **8**. The geometry structure optimized by DFT calculations using this spin state is consistent with the single-crystal structure.

Reviewer 2 wrote: “The N3-N4-B1 angle is 124.01(17)°, suggesting sp^2 hybridization of the N_β atom with a dative coordinated boron atom” This time the NNB angle is more acute than in **6**. Can the authors comment on that?

Response: Thanks for your suggestion. As we mentioned above, the N4–B1 bond (N4–B1 bond length: 1.423(3) Å) of complex **6** is more like a covalent bond with multiple B–N bonding, while the N4–B1 bond (N4–B1 bond length: 1.618(3) Å) of complex **8** is a dative bond located at one of the sp^2 orbital of N_β atom. Therefore, the N3-N4-B1 angle of complex **8** is closer to 120° than complex **6**.

Reviewer 2 wrote: I have doubt regarding the formulation of **10** that could not be isolated in the present work. On which basis do the authors propose a two-fold adduct against a mono-adduct? Did they run computations on **1** and $[\text{Cp}^*(\text{I}^t\text{Pr}_2\text{Me}_2)\text{Cr}(\text{NNBEt}_3)(\text{N}_2)]\text{K}$ and analyze the trend in the calculated N-N stretching frequencies? Looking at the literature concerning Lewis adducts of N_2 complexes, I found an interesting recent study by Escomel *et al.* (*Chem. Sci.* **15**, 11321–11336 (2024)) that shows that coordination of one $\text{Al}(\text{C}_6\text{F}_5)_3$ to *cis*- $[\text{W}(\text{N}_2)_2\text{L}_4]$ lowers the energy of the stretch of the bridging N_2 but increases the one of the terminal N_2 . However, coordination of two $\text{Al}(\text{C}_6\text{F}_5)_3$ to *cis*- $[\text{W}(\text{N}_2)_2\text{L}_4]$ results in asymmetric and symmetric (coupled) N-N stretches that are both red-shifted when compared to those of naked *cis*- $[\text{W}(\text{N}_2)_2\text{L}_4]$. The phenomenon that is observed by the authors with IR spectroscopy seems, in view other people’s work, characteristic of a mono-adduct, namely $[\text{Cp}^*(\text{I}^t\text{Pr}_2\text{Me}_2)\text{Cr}(\text{NNBEt}_3)(\text{N}_2)]\text{K}$, contrary to the explanation note they give in the supporting information. The presence of bands characteristic of **1** indeed point to an equilibrium in their case. Addition of more BEt_3 (or AlMe_3) shifts accordingly the equilibrium towards **10** and it would be welcome to attempt pushing the equilibrium towards a quantitative formation of **10** using a huge excess of the LA. The characterization of **10** thus deserves more investigations.

Response: We thank the reviewer for the valuable suggestions. We performed frequency calculations using the TPSSh/def-TZVP level of theory. We applied a frequency scaling factor of 0.96. The computed vibrational frequencies for the dinitrogen ligand in **1-crypt** were 1878 cm^{-1} and 1779 cm^{-1} , which closely matched the experimental values. Upon coordination with two equivalents of BEt_3 , the calculated IR vibrational peaks shifted to 1917 cm^{-1} and 1752 cm^{-1} , also aligning well with the experimental data, supporting the formation of a bis-coordinated species upon Lewis acid coordination.

Additionally, we calculated the ΔG values for the coordination of one and two equivalents of BEt_3 to the **1-crypt** anion in THF at -30 °C, obtaining ΔG values of +0.7 kcal/mol for one equivalent and -0.8 kcal/mol for two equivalents. This supports the reviewer's hypothesis that an equilibrium exists in solution, favoring the coordination of two equivalents of BEt_3 . However, the driving force is relatively small and is strongly dependent on temperature and concentration, making it challenging to isolate crystals of complex **10-BEt₃**.

Moreover, we continued to add BEt_3 to the reaction system, from 1.0 equiv. to 20.0 equiv. in the range of $-60\text{ }^\circ\text{C}$ to $-100\text{ }^\circ\text{C}$. The peaks at $1738, 1900\text{ cm}^{-1}$ for the two-fold adduct **10-BE t_3** and at $1780, 1869\text{ cm}^{-1}$ for **1-crypt** are the primary nitrogen-related signals, with no additional N_2 -related peaks observed. All these findings further confirm that bis-coordination is more favorable in the solution.

In addition, we carried out *in situ* ^{15}N NMR experiment for the reaction of **1-crypt** and 2.0 equiv. BEt_3 . Two new peaks (385 and 416 ppm), distinct from **1-crypt** (404 and 413 ppm), were observed in the *in situ* ^{15}N NMR spectra of **1-crypt** with 2.0 equiv. BEt_3 , suggesting that double coordination with a symmetrical structure is more likely than mono coordination with an asymmetrical structure.

Figure S22 ^{15}N NMR spectrum of *in situ* reaction of **1-crypt** and 2.0 equiv. BEt_3 in THF-d^8 at room temperature

Reviewer 2 wrote: For future work, the authors should also consider strong Lewis acids such as the perfluoroaryl-substituted ones.

Response: We rechecked the reactions between **1-crypt** and BPh_3 or $\text{B}(\text{C}_6\text{F}_5)_3$. For BPh_3 , the newly formed peaks $1733\text{ cm}^{-1}, 1912\text{ cm}^{-1}$ indicates that the SET process occurred between **1-crypt** and BPh_3 . For $\text{B}(\text{C}_6\text{F}_5)_3$, there are no N_2 unit related peaks, indicates other side reactions occurred.

Overall, we did not obtain clear results when using these aryl-substituted boron Lewis acids.

Reviewer 2 wrote: Have the authors attempted to add two equivalents of MeOTf on **10-AIME₃** to assess if the *bis*-methylated *bis*diazenido product or even the dimethyl hydrazido compound can form?

Response: We carried out the reaction of **1-K** with 2.0 equiv. BEt₃ and 2.0 equiv. MeOTf, and we isolated comparable amount of *mono*-methylated product **8** instead of *bis*-methylated *bis*diazenido products **8A**, **8B** or the dimethyl hydrazido compound **8C**.

Reviewer 2 wrote: “reducing the reducibility of Cr center” is counter-intuitive, or the authors formulated their idea in an unclear manner. Indeed, if the LA is pulling electron density from the metal center, it should be either easier to reduce (so diminished reducibility) or more difficult to oxidize. I would rephrase in “reducing the propensity of Cr to get oxidized”

Response: Revised.

Reviewer 2 wrote: “Additionally, the increased steric hindrance of BEt₃ effectively prevents undesired reactions between MeOTf and the Cr center” Is steric hindrance really the key here (I would not qualify BEt₃ of a sterically hindered borane), or the effect of LA coordination on the N₂ ligand? Can the authors provide elements of comparison (BMe₃, BF₃, BiPr₃) for this specific reaction?

Response: We thank the reviewer for the valuable suggestions. If the BF₃ was used to construct N–C bond instead of BEt₃, we cannot obtain corresponding products. So we guess that BEt₃ can prevent side reaction between MeOTf and Cr center to some extent. We agree with you that LA coordination cannot be ignored and is the main reason to reduce the propensity of Cr

to get oxidized.

The main text was revised accordingly.

Reviewer 2 wrote: “once the N-C bond is formed, BEt₃ does not dissociate” have the authors tried if treatment with a Lewis base (e.g., DMAP) allows borane dissociation?

Response: We thank the reviewer for the valuable suggestions. We carried out the reaction between **8** and 1.0 equiv. DMAP. The green color of **8** in THF didn't change and we isolated the main component is **8**, indicating there is no reaction between **8** and DMAP.

Reviewer 2 wrote: Similar to Szymczak (ref. 51), could the authors expand their exemplification and demonstrate the value of Lewis acid adjunction by comparing the reactivity of **1** vs **1**·LA with proton sources?

Response: We thank the reviewer for the valuable suggestions. Unfortunately, our system does not readily allow for the formation of N-H bonds. We consistently get the byproduct I'Pr₂Me₂·HX (X = OTf, BAr^F₄), which results from the protonation of the ligand.

Reviewer 2 wrote: “In an nitrogen atmosphere glovebox” should read “In a nitrogen atmosphere glovebox”

Response: Revised.

Response to Comments of Reviewer 3

Reviewer #3 (Remarks to the Author):

The manuscript of Wei, Xi and coworkers presents a new approach to electrophilic functionalization of TM-bound N₂ molecules, by the initial coordination of Lewis acids, followed by the electrophile. This is demonstrated on a paramagnetic CpCr(NHC) framework, and results in the construction of bonds from nitrogen to a range of p-block elements.

To me, the work is very nice and the N-E bonds are impressive, but it seems a little incomplete, as it is really just the first steps in a more full N₂ functionalization / dissociation. I would be OK with accepting the manuscript with its current chemistry, but further functionalization of the N₂ unit would give a more complete picture, while also showing something closer to real utility. Adding something like that would be a great improvement, and would make the manuscript a better fit in a prestigious journal such as this.

Response: We thank the reviewer for valuable comments. Based on the suggestions provided by you and other reviewers, we have significantly improved the manuscript by adding new content, which has substantially enhanced the overall quality of the work.

In particular, we have discovered that BEt₃ can be removed by IMe₄ (IMe₄ = 1,3-dimethyl-4,5-dimethylimidazol-2-ylidene), leading to a rare and novel transition of the [NNMe] moiety from end-on to side-on coordination. This represents an interesting and previously unexplored

transformation. This part was added in the main text as Fig. 5. Meanwhile, the coordinated BEt_3 in **8** can be replaced by the more acidic AlMe_3 , forming **9**, with no significant structural changes observed after Lewis acid substitution (Fig. 5).

Fig. 5 | Further transformation of complex 8. Selected bond lengths [Å] and angles [deg] of **9**: Cr1–N3 1.694(2), N3–N4 1.286(3), N4–Al1 1.944(2), N3–N4–Al1 121.28(16); **11**: N1–N2 1.389(2), N3–N4 1.387(2), N1–N2–C1 119.39(15), N3–N4–C2 119.42(17).

The transformation of N_2 has always been our goal, we actually have tried our best to explore numerous possibilities for further converting these products. However, the reactions of **2b** with CO_2 , $t\text{BuNCO}$, PhSiH_3 , $t\text{Bu}_3\text{SnH}$, DIBAL-H or KC_8 , as well as the anticipated [3+2] cycloaddition of **4** with CO_2 , $t\text{BuNCO}$, or $\text{RC}\equiv\text{CR}$ ($\text{R} = \text{Me}, \text{COOMe}$) have not yet succeeded.

Reviewer 3 wrote: Fig 1 is very confusing - the variable E is defined underneath the word "diazenido", but then E^+ is defined just to the right of that, with different atoms. This is duplication, but also they don't match entirely, which is confusing. LA should also be defined as Lewis acid.

Response: Revised.

Fig. 1 | N-E bond formation via associative pathways. a Challenges for diverse N-E bonds formation via diazenido complexes. b N-E bond formation beyond N-H bond promoted by Lewis acids. LA = Lewis acid.

Reviewer 3 wrote: Fig 2 - above the arrow, "E⁺" is stated as a reagent. However, E⁺ itself is not added to the reaction. The actual reagent should be specified (E–Cl or E–X, where E = SiMe₃, ... ; X = ...). The same goes for Fig 4.

Response: Revised.

Reviewer 3 wrote: The synthesis and structure of 7 should be shown somewhere in a figure.

Response: Thanks, we added the synthesis and structure of 7 in Fig. 2.

Reviewer 3 wrote: The manuscript needs to be more explicit about whether the compounds are paramagnetic or not. For some complexes it is stated, for others it is missing (e.g. 3 and 4).

Response: We thank the reviewer for the valuable suggestions. We added our comments in the revised manuscript.

Reviewer 3 wrote: I assume the paramagnetism is also the reason for the missing C, B, F and N NMR data, but it would be nice if the authors would state this clearly somewhere in the manuscript, and if they attempted to record spectra for these other nuclei.

Response: We thank the reviewer for the valuable suggestions. Actually, we tested ²⁷Al NMR for 2a, ¹¹B NMR for 2b, ¹¹B NMR for 3, ¹¹B NMR and ³¹P NMR for 4, ¹¹B NMR for 6, ¹¹B NMR for 8, but only the ¹¹B NMR of 8 shows signal at 21.41 ppm, other signals could not be detected even at high concentrations, presumably as a consequence of paramagnetic or the presence of the quadrupolar ¹¹B nucleus. We have stated these in the revised Supporting Information.

Reviewer 3 wrote: Conclusion: "offering a viable approach for isolating N–B, N–Ge, N–P bond-containing value-added compounds in 3d metals for the first time" - this is a bit of an exaggeration. I wouldn't consider it a viable approach to isolating value-added compounds, because no real value-added compounds were prepared in the work. However, it could be a "potential" approach to such compounds.

Response: We thank the reviewer for the valuable suggestions. We revised the sentence.

Response to Comments of Reviewer 4

Reviewer #4 (Remarks to the Author):

Point-by-Point Response to Comments of Reviewers

Response to Comments of Reviewer 1

Reviewer #1 (Remarks to the Author):

The authors have made significant efforts to enhance the theoretical analysis and have addressed all my concerns thoroughly and satisfactorily. I am pleased to recommend this manuscript for publication and congratulate the authors for their work.

Response: We thank the reviewer for valuable comments on this work and strongly support.

Response to Comments of Reviewer 2

Reviewer #2 (Remarks to the Author):

The authors have revised their manuscript and addressed most of the points raised by the reviewers, especially by providing more computational and experimental data, greatly improving its quality. Initially, I was skeptical about what the authors claimed for the formulation of **10-BEt₃** as a bis-adduct based on their IR data. However, they successfully solidified their characterisation and their claim by providing theoretical N-N IR stretches and energy profiles, and experiments with an excess of the LA. Authors also proposed the design of a new species, compound **11**, by extrusion of BEt₃ in complex **8** with a Lewis base (IME₄). This species is quite uncommon (dimerization phenomenon and leaving of the NHC ligand from Cr) and is a nice example of how Lewis acids may help access original N₂-activation complexes. If my initial feeling was that the manuscript was more fitted for a specialised journal, I acknowledge that the authors have seriously taken into account the criticisms of all reviewers and strengthened substantially the chemistry described here. My appreciation of this paper in terms of novelty does not change, but if laudatory reports come back from the two other reviewers and that the editor gives acceptance as final decision, I congratulate to the authors for having this elegant work published.

Response: We thank the reviewer for valuable comments on this work and strongly support.

Response to Comments of Reviewer 3

Reviewer #3 (Remarks to the Author):

The authors have impressively tackled the Herculean task of addressing all of the 30+ comments of referees 1&2, although I will leave it up to these referees to say how well the points were addressed. My comments were dealt with satisfactorily, and the addition of further reaction chemistry adds some weight to the work. I am happy to recommend acceptance, with two minor further points:

Response: We thank the reviewer for valuable comments on this work and strongly support.

Reviewer 3 wrote: Complex **11** seems to be quite novel to me. Is there anything like it in the literature? Is it the first example of a complex with side-on-bound alkylated or substituted N₂ ligands? This could be explored and, if it is, mentioned.

Response: We thank the reviewer for valuable comments. Although such examples are very rare, there are several reports in the literature describe the side-on-coordinated alkylated or substituted N₂ ligands (*J. Am. Chem. Soc.* **2004**, *126*, 9480-9481; *J. Am. Chem. Soc.* **2007**, *129*, 12690-12692; *J. Am. Chem. Soc.* **2024**, *146*, 26, 17624–17628.). **Our study here represents**

the first example of a transformation from an end-on to a side-on alkylated or substituted N₂ ligand.

Reviewer 3 wrote: Complex **11** is not named in the text.

Response: Revised.

Response to Comments of Reviewer 4

Reviewer #4 (Remarks to the Author):

This article by Yin *et al.* studies (by spectroscopic, crystallographic, and computational methods) the influence of Lewis acids (LA) to the distal N of Cr-N₂ complexes on derivatization of the N₂ ligand into a silyl-, germyl-, boryl, aluminyl-, phosphanyl- or methyl-diazenido one, by reaction with main-group electrophiles. Their results points to a positive effect of such association, granting isolation of a family of N₂-derived diazenido compounds, in particular the methyl-substituted ones that could not be prepared without LA association. This manuscript also highlights two reactional approaches to get these fleeting diazenido compounds: i) Reaction of [Cp*(L)Cr(N₂)₂]K with the LA to trigger pre-activation of the terminal nitrogen of the N₂ unit which is then subjected to electrophilic functionalization with a Si, Ge, C, or P-based electrophilic agent. ii) The reverse approach, which entails the *in situ* generation of the reactive fleeting diazenido intermediates - [Cp*(L)Cr(μ-N₂-E)] - that is then treated with the LA to isolate and trap the resulting species. This set of results is indeed important in the context of N fixation beyond ammonia and further establishes transition metal complexes as the best way to prepare value-added nitrogenous compounds from N₂. Formation of diazenido compounds from N₂ is not new, but, as the authors underline, difficult with 3d metals. Facilitating this by introduction of LA coordination is clever. Such a strategy builds on the seminal work from the Szymczak group demonstrating that for iron-N₂ complexes, adjunction of an LA that coordinates to the terminal N of the N₂ ligand enhances back-donation from the metal, making the N₂ ligand more electron-rich at the terminal N and allowing selective electrophilic functionalization thereof, instead of reactivity at the metal center. This work is technically sound, the species are well characterized and I congratulate the authors for managing handling of such low-valent Cr-N₂ complexes that are very challenging sensitive species, some successfully trapped and isolated at low temperature (down to -100°). They are comprehensively characterized by a wide range of tools. In my opinion, this work represents a nice follow-up of this group's chemistry on Cr-N₂ complexes, but lacks novelty in terms of N₂ transformation. It is rather an elegant combination of different known reactivity of N₂ complexes that lead the authors to solve a problem on a reaction (the treatment of **1** or related complexes with electrophiles) they have already reported on (JACS 2019, JACS 2023, JACS 2023). Therefore, I consider it would be more suited for a specialized journal. Would they choose to submit elsewhere or would the other reviewers be less severe so that the manuscript has chances to make its way through, I invite the authors to consider addressing the following points which, I believe, would greatly improve the quality of the manuscript and lift some interrogations.

• Introduction:

- “Over the past six decades, the N-H bond formation facilitated by transition metals via associative and dissociative pathways has been extensively studied” It is not clear what the authors mean by associative and dissociative pathways. Do they refer to alternating and distal pathways? When saying transition metals, do they only consider soluble complexes? “N-H bond formation” is vague and could refer to a lot of transformations, while given the content of the article it is likely that the authors refer specifically to N₂ and N₂ derived ligands. Please rephrase.
- “the isolation of diazenido compounds containing N-C, N-Si, N-Ge, and N-P bonds” Reference 41 is given to support the words, the part 1 of Wiberg's review about azenes is more appropriate (part 2 is cited): Wiberg, N. Silyl, Germyl, and Stannyl Derivatives of Azenes, NnHn: Part I. Derivatives of Diazene, N₂H₂. in *Advances in Organometallic Chemistry* (eds. Stone, F. G. A. & West, R.) vol. 23 131–191 (Academic Press, 1984). A more recent example of isolation of poorly stable silyldiazene adducts of Lewis acids is also worth mentioning: Reiß, F., Schulz, A. & Villinger, A. Synthesis, Structure, and Reactivity of Diazene Adducts: Isolation of iso-Diazene Stabilized as a Borane Adduct. *Chem. Eur. J.* 20, 11800–11811 (2014).
- “Using both transition-metal complexes and Lewis acids (LA) to co-activate N₂ thus represents a promising approach” Beyond the references given by the authors, other chemists have reviewed this topic: Ruddy, A. J., Ould, D. M. C., Newman, P. D. & Melen, R. L. Push and pull: the potential role of boron in N₂ activation. *Dalton Trans.* 47, 10377–10381 (2018); Simonneau, A. & Etienne, M. Enhanced Activation of Coordinated Dinitrogen with p-Block Lewis Acids. *Chem. Eur. J.* 24, 12458–12463 (2018). Some recent development should also be added: Jori, N. et al. Iron promoted end-on dinitrogen-bridging in heterobimetallic complexes of uranium and lanthanides. *Chem. Sci.* 15, 6842–6852 (2024); Escomel, L. et al. Coordination of Al(C₆F₅)₃ vs. B(C₆F₅)₃ on group 6 end-on dinitrogen complexes: chemical and structural divergences. *Chem. Sci.* 15, 11321–11336 (2024).
- “More accessible conditions for N-C bond formation were achieved by inhibiting undesired side reactions at the Cr(0) center” It would be interesting to note here that for an iron-phosphine-N₂ complex, electrophiles such as H⁺ or Me⁺ reacted at the metal while Si⁺ electrophiles reacted at the N₂ ligand, similar to the authors' Cr complexes: Field, L. D., Hazari, N. & Li, H. L. Nitrogen Fixation Revisited on Iron(0) Dinitrogen Phosphine Complexes. *Inorg. Chem.* 54, 4768–4776 (2015).

• Results and discussion:

- “the N_β atom in diazenido intermediates is already sp²-hybridized with a lone pair of electrons” It is probably pertinent to cite here the reviews by Sutton and Dilworth about diazenido compounds: Dilworth, J. R. Diazene, diazenido, isodiazene and hydrazido complexes. *Coord. Chem. Rev.* 330, 53–94 (2017); Sutton, Derek. Organometallic diazo compounds. *Chem. Rev.* 93, 995–1022 (1993).

- “Both **2a** and **2b** are paramagnetic and have a solution magnetic moment of 2.4(1) μB and 2.7(1) μB at 296 K, respectively” Are these values consistent with the expected spin state of Cr(II) within a strong field, half-sandwich coordination?
- “The reaction between **1-K** and common B-based electrophiles (Cy_2BCl , Ph_2BCl , Mes_2BF) fail to construct N–B bonds, likely due to insufficient steric hindrance” The meaning of this sentence is not clear. Cy and Mes group are rather bulky. The steric hindrance should play a role if a N–B bond is formed, yet the authors declare that this was not the case. Could the authors clarify whether they could evidence N–B bond formation (e.g., by IR) and that the thus-formed boryldiazenido compound is unstable, or if decomposition occurs without N–B bond formation and in that case the argument of steric hindrance is not adequate?
- “and less electrophilicity B-based electrophile” should read “and less electrophilic B-based electrophile”
- “and the N3–N4–B1 angle (138.56(19)°) suggests an sp^2 hybridized geometry for the N_β atom.” This angle significantly deviates from the 120° expected for an ideal sp^2 -hybridized p-block atom. Can the authors explain this and compare whether this is the case for other boryldiazenido compounds?
- surprisingly, the authors do not comment on the success or failure of applying their second strategy (adding LA to **1** then E^+) to the problematic boron halides Cy_2BCl , Ph_2BCl and Mes_2BF . Have the authors tried?
- can the authors assign N–N stretching frequency in their IR characterizations of the diazenido complexes (or at least part of them) with the help of ^{15}N -labeling?
- “This phenomenon, reminiscent of Lewis acids mimicking nitrogenase” is in my opinion too vague to express what the authors mean here. It should be rephrased in that sense: “The lowering of the stretching frequency for the N_2 ligand is indicative of a diminished bond order and greater polarization. This is reminiscent of the influence of acidic residues within the active site of the nitrogenase on Fe-bound N_2 , further polarizing it and presumably assisting protonation.” Reference to the work of Szymczak (ref. 51) must be made here.
- “Complex **8** is paramagnetic and has a solution magnetic moment of 3.2(1) μB at 296 K” Same as above, does this value allows to extract a meaningful spin state?
- “The N3–N4–B1 angle is 124.01(17)°, suggesting sp^2 hybridization of the N_β atom with a dative coordinated boron atom” This time the NNB angle is more acute than in **6**. Can the authors comment on that?
- I have doubt regarding the formulation of **10** that could not be isolated in the present work. On which basis do the authors propose a two-fold adduct against a mono-adduct? Did they run computations on **1** and $[\text{Cp}^*(\text{tPr}_2\text{Me}_2)\text{Cr}(\text{NNBET}_3)(\text{N}_2)]\text{K}$ and analyze the trend in the calculated N–N stretching frequencies? Looking at the literature concerning Lewis adducts of N_2 complexes, I found an interesting recent study by Escomel *et al.* (*Chem. Sci.* **15**, 11321–11336 (2024)) that shows that coordination of one $\text{Al}(\text{C}_6\text{F}_5)_3$ to *cis*- $[\text{W}(\text{N}_2)_2\text{L}_4]$ lowers the energy of the stretch of the bridging N_2 but increases the one of the terminal N_2 . However, coordination of two $\text{Al}(\text{C}_6\text{F}_5)_3$ to *cis*- $[\text{W}(\text{N}_2)_2\text{L}_4]$ results in asymmetric and symmetric (coupled) N–N stretches that are both red-shifted when compared to those of naked *cis*- $[\text{W}(\text{N}_2)_2\text{L}_4]$. The phenomenon that is observed by the authors with IR spectroscopy seems, in view other people’s work, characteristic of a mono-adduct, namely $[\text{Cp}^*(\text{tPr}_2\text{Me}_2)\text{Cr}(\text{NNBET}_3)(\text{N}_2)]\text{K}$, contrary to the explanation note they give in the supporting information. The presence of bands characteristic of **1** indeed point to an equilibrium in their case. Addition of more BET_3 (or AlMe_3) shifts accordingly the equilibrium towards **10** and it would be welcome to attempt pushing the equilibrium towards a quantitative formation of **10** using a huge excess of the LA. The characterization of **10** thus deserves more investigations.
- For future work, the authors should also consider strong Lewis acids such as the perfluoroaryl-substituted ones.
- Have the authors attempted to add two equivalents of MeOTf on **10-AlMe₃** to assess if the *bis*-methylated *bis*-diazenido product or even the dimethyl hydrazido compound can form?
- “reducing the reducibility of Cr center” is counter-intuitive, or the authors formulated their idea in an unclear manner. Indeed, if the LA is pulling electron density from the metal center, it should be either easier to reduce (so diminished reducibility) or more difficult to oxidize. I would rephrase in “reducing the propensity of Cr to get oxidized”
- “Additionally, the increased steric hindrance of BET_3 effectively prevents undesired reactions between MeOTf and the Cr center” Is steric hindrance really the key here (I would not qualify BET_3 of a sterically hindered borane), or the effect of LA coordination on the N_2 ligand? Can the authors provide elements of comparison (BMe_3 , BF_3 , BiPr_3) for this specific reaction?
- “once the N–C bond is formed, BET_3 does not dissociate” have the authors tried if treatment with a Lewis base (e.g., DMAP) allows borane dissociation?
- Similar to Szymczak (ref. 51), could the authors expand their exemplification and demonstrate the value of Lewis acid adjunction by comparing the reactivity of **1** vs **1**·LA with proton sources?
- **Methods:**
 - “In an nitrogen atmosphere glovebox” should read “In a nitrogen atmosphere glovebox”